# Cortical ignition dynamics is tightly linked to the core organisation of the human connectome

**Samy Castro** [1,2‡], **Wael El-Deredy** [3], **Demian Battaglia** [4‡*], **Patricio Orio** [1,5‡*]

**1** Centro Interdisciplinario de Neurociencias de Valparaíso, Universidad de Valparaíso, Valparaíso, Chile,
**2** Programa de Doctorado en Ciencias, mención Neurociencia, Universidad de Valparaíso, Valparaíso, Chile,
**3** Centro de Investigación y Desarrollo en Ingeniería en Salud, Universidad de Valparaíso, Valparaíso, Chile,
**4** Aix-Marseille Université, Institut de Neurosciences des Systèmes, INSERM UMR 1106, Marseille, France,
**5** Instituto de Neurociencias, Facultad de Ciencias, Universidad de Valparaíso, Valparaíso, Chile

‡ This author has first authorship on this work. DB and PO share last authorship on this work.
* demian.battaglia@univ-amu.fr (DB); patricio.orio@uv.cl (PO)

**Data Availability Statement:** All relevant data are within the manuscript and its Supporting Information files.

**Funding:** This work was supported by Fondecyt Grants 1181076 (to PO) and 1201822 (to WeD),

## Abstract

The capability of cortical regions to flexibly sustain an "ignited" state of activity has been discussed in relation to conscious perception or hierarchical information processing. Here, we investigate how the intrinsic propensity of different regions to get ignited is determined by the specific topological organisation of the structural connectome. More specifically, we simulated the resting-state dynamics of mean-field whole-brain models and assessed how dynamic multistability and ignition differ between a reference model embedding a realistic human connectome, and alternative models based on a variety of randomised connectome ensembles. We found that the strength of global excitation needed to first trigger ignition in a subset of regions is substantially smaller for the model embedding the empirical human connectome. Furthermore, when increasing the strength of excitation, the propagation of ignition outside of this initial core–which is able to self-sustain its high activity–is way more gradual than for any of the randomised connectomes, allowing for graded control of the number of ignited regions. We explain both these assets in terms of the exceptional weighted core-shell organisation of the empirical connectome, speculating that this topology of human structural connectivity may be attuned to support enhanced ignition dynamics.

## Author summary

The activity of the cortex in mammals constantly fluctuates in relation to cognitive tasks, but also during rest. The ability of brain regions to display ignition, a fast transition from low to high activity is central for the emergence of conscious perception and decision making. Here, using a biophysically inspired model of cortical activity, we show how the structural organization of human cortex supports and constrains the rise of this ignited dynamics in spontaneous cortical activity. We found that the weighted core-shell organization of the human connectome allows for a uniquely graded ignition. This graded ignition implies a smooth control of the ignition in cortical areas tuned by the global

the Advanced Center for Electrical and Electronic Engineering (FB0008 ANID, Chile) and the supercomputing infrastructure of the NLHPC (ECM-02). The Centro Interdisciplinario de Neurociencia de Valparaíso (CINV) is a Millenium Institute supported by the Millennium Scientific Initiative (ANID). SC was funded by Beca Doctorado Nacional ANID 21140603 and by Programa de Doctorado en Ciencias, mención Neurociencia, Universidad de Valparaíso. DB acknowledges support from the EU Innovative Training Network "i-CONN" (H2020 ITN 859937). The funders had no role in study design, data collection and analysis, decision to publish, or preparation of the manuscript.

**Competing interests:** The authors have declared that no competing interests exist.

excitability. The smooth control cannot be replicated by surrogate connectomes, even though they conserve key local or global network properties. Indeed, ignition in the human cortex is first triggered on the strongest and most densely interconnected cortical areas–the "*ignition core*"–, emerging at the lowest global excitability value compared to surrogate connectomes. Finally, we suggest developmental and evolutionary constraints of the mesoscale organization that support this enhanced ignition dynamics in cortical activity.

## Introduction

Human (*H. sapiens*) cognition relies on the coordinated recruitment of distributed brain-wide networks, which are flexibly reconfigured depending on external context and internal brain state [1]. Even at rest, functional connectivity between brain regions is restless, transiently visiting a multiplicity of metastable configurations [2,3], which are reminiscent of cognitive networks evoked during specific tasks [4]. Such dynamic functional connectivity has been considered to stem from the complex collective dynamics of brain networks [5], which is necessarily shaped by the underlying structural connectome [6–9]. In particular, based on theoretical neuroscience insight [10,11], one expects that a richly structured "chronnectome"–i.e., the repertoire of functional connectivity states observed at different times [12]–arises when the noise-driven dynamics of brain networks can sample an equally rich "dynome" [13], i.e. a repertoire of multi-stable dynamical states [14,15] or characteristic transient fluctuation modes [16,17].

Particularly important is the possibility for brain regions to develop bistability between a baseline state at low firing rate activity and a second "ignited" state in which the firing rate is substantially higher, often associated with a functional role in working memory or input integration [18,19]. In the language of statistical mechanics, multistability is due to a discontinuous phase transition associated to hysteresis. Once a region first enters into an ignited state, as an effect of input bias or spontaneous fluctuations (e.g. a stimulus or a top-down signal), this early ignition can then propagate to neighbouring regions, eventually recruiting them as well into an ignited network core. Related processes may allow the access of a perceptual stimulus to the conscious workspace [20,21] or mediate cross-scale integration of information processing by hierarchical brain networks [8,9,22]. Remarkably, several studies about mean-field computational models of the resting state–which were not intended to explore cortical ignition directly–have also consistently reported that the best fit between simulated and empirical functional connectivity is found in a critical range of global coupling where switching between ignited and not-ignited network states is possible [15,23].

Growing experimental [20] and modelling [24] evidence stresses how cortical ignition is non-linear in nature, with regions able to get ignited only if the inputs they receive–external, but also, notably, recurrent–rise above a minimum threshold. Whether this threshold is crossed or not depends on a variety of factors, such as the number of neighbouring regions and the strength of incoming connections, but also the activity state of the neighbouring regions themselves [22], influenced on its turn by the network collective state. For this reason, it is difficult to disentangle the relative contributions of the structural connectome or network dynamics in determining the propensity of different regions to sustain a high-firing rate state, at earlier or later stages of the ignition cascade. The human connectome is associated with specific distributions of the local organisation, such as node *degrees* (i.e. the number of neighbouring regions) or node *in-* or *out-strengths* (i.e. the sum of the weights of incoming or outgoing

connections), as well as of global organisation such as *small-worldness* [25,26]. It is not clear a priori how these different specific levels of organisation of the connectome influence the ignition behaviour of mean-field models built on them.

Here, we systematically explore the factors favouring ignition–at both the connectome and dynome levels–, by using a mean-field whole-brain modelling approach. Focusing on the intrinsic tendency to ignition (i.e. in the absence of external stimuli), we study how the repertoire of possible spontaneous states of activity of the model evolves as a function of the strength of effective inter-regional coupling gain. Confirming the previously mentioned results [15,23], we identify a range of effective inter-regional coupling gain $G$ in which the network dynamics is multi-stable, delimited the *ignition point* $G_-$, below which no region gets ignited; and the *flaring point* $G_+$, above which nearly all regions will be ignited regardless of initial conditions. We find that the existence of multistability is not exclusive to the used connectome since both the ignition and the flaring points arise even in models embedding a variety of different random connectivity matrices. Nevertheless, we also find that the ignition dynamics observed when using an empirical DSI-derived (Diffusion Spectrum Imaging) connectome [27] has some special features that are remarkably deviating from randomised models.

First, ignition is highly facilitated, as revealed by the fact that the ignition point $G_-$ arises at substantially smaller values (i.e. a weaker inter-regional excitation is needed to first sustain an ignited core) for the empirical connectome than for randomised connectomes with a preserved degree or weight distributions or small-worldness. Second, the cortical ignition dynamics is particularly graded for the empirical connectome. Indeed, at the ignition point $G_-$ itself, the subset of regions getting first ignited is smaller and more compact than for any of the randomised connectomes. Furthermore, the recruitment of additional regions into the ignited subset is particularly gradual when increasing inter-regional coupling. We thus speculate that the empirically measured human connectome may have been attuned through evolution or development to give rise to enhanced cortical ignition features.

## Results

### Mean-field model of whole-brain resting-state dynamics

In a computational modelling framework, we can re-define ignition as a transition from a state of low activity to a high activity, where each brain region is modelled as a neural mass system and its activity is described by a collective rate variable, $R_i$ [28]. Following Deco et al. [23] and Hansen et al. [15], we chose here to use a reduced Wong-Wang model neural mass. Obtained from the simplification of a local dynamics model initially meant to capture bistable behaviour in working memory and decision making [19], such a model can develop two types of steady-state dynamics, one with low firing rate ("baseline state") and one with high firing rate ("ignited state") as a function of its parameters (controlling local excitability) and the received input. In the context of this study, we use the term "ignition" to refer to switching from the low firing rate baseline state to the higher firing rate state, associated to the existence of dynamic bistability. As shown in Fig 1A, in a suitable range of conditions (see Fig A in S1 File for more details), the Wong-Wang model of an isolated region can already develop intrinsic bistability between a low activity baseline and a high activity ignited state, making such a model particularly suitable for our computational investigation of cortical ignition dynamics.

When moving from regional to whole-brain network dynamics, we can expect, in agreement with several authors [6,15,20,29,30] that the spontaneous ignition dynamics and state shifting in different regions will be shaped by the underlying structural connectivity (SC) included in the model. In this study, we used as reference cortical connectome a connectivity matrix mediated from Hagmann et al. [27], based on an average of 5 right-hand male subjects

 

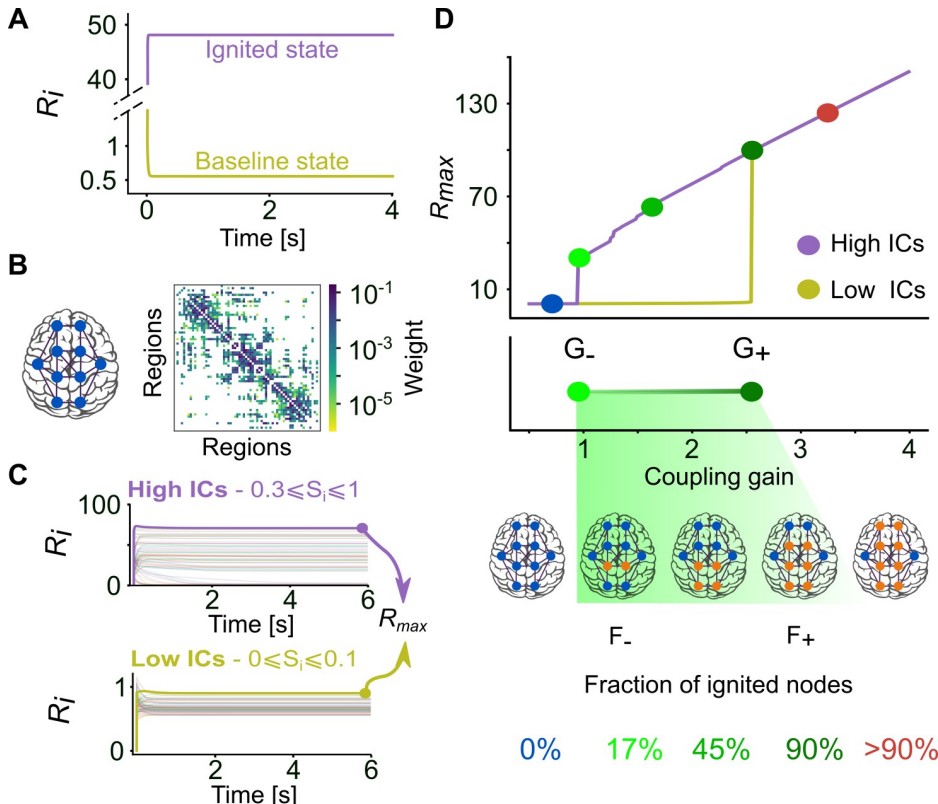

**Fig 1. Ignition state in a mean-field whole-brain model of the human cortical connectome.** (A) Steady-state mean firing rate ($R_i$) dynamics of an isolated cortical area, showing both the baseline state (*yellow*) and the ignited state (*purple*) of activity in the mean-field model. (B) Structural connectivity (SC) matrix of the averaged five male subjects [27]. The colour scale shows the coupling weight between cortical areas in log scale. (C) Activation level of each cortical area at $G = 1.8$ and different initial conditions (ICs). Top, the ignited network state emerges from High ICs ($0.3 \leq S_i \leq 1$). Bottom, the baseline network state arises from Low ICs ($0 \leq S_i \leq 0.1$). $R_{max}$ is the highest steady-state value among cortical areas and is used to define the network activity level. (D) Top, network activity level using the human SC as a function of coupling gain ($G$), starting either from Low (*yellow*) or High (*purple*) ICs. Middle, bistable range of ignition in the model, starting at ignition point G- (0.945, *light green circle*) and ending at the flaring point G+ (2.545, *dark green circle*). Bottom, fraction of ignited nodes, $F_{ignited}$ (threshold $R_i > 5$), increasing from F-~17% (11 nodes) in G- to F+~90% (59 nodes) in G+. The coupling range was $0.5 \leq G \leq 4$, with steps of $\Delta G = 0.01$. The parameters of the mean-field model are $I_0 = 0.322$ and $\omega = 1$ in **A** and $I_0 = 0.3$ and $\omega = 0.9$ in **C** and **D**.

diffusion MRI (Magnetic Resonance Imaging) data. This connectome is parcellated following the Desikan-Killany atlas [31], which has 66 cortical areas (33 per hemisphere) wired by 1148 cortico-cortical connections (Fig 1B, see Table A in S1 File for a list of regions). When inserting a Wong-Wang neural mass at each node of the connectome, we could observe diverse activation levels in different regions as a function of their specific connectivity neighbourhood (all regional parameters were otherwise identical). These heterogeneous activation levels could be distinguished into low or high activation ranges, with a clear gap separating the two (Fig 1C). Thus, the capability for bistable activation of the isolated regional model was maintained as well when embedding the neural masses in a wider connectome. The actual fraction of regions that were entering an ignited (or bistable) state depended eventually on the global strength of long-range connections $G$ as well as the initial conditions (ICs) for the network activity (High ICs or Low ICs).

The connectome matrix mediated from [27] sets the relative strength of different connections but not the absolute strength of these connections. The connectome matrix is multiplied

by a constant coupling gain multiplier $G$ which sets the strength of influence of long-range cortico-cortical inputs on regional dynamics with respect to local recurrent connectivity (summarized in the local neural mass parameters). We plot in Fig 1D the maximum mean firing rate, $R_{max}$, across all regions as a function of growing $G$. When $G$ is too small ($G<G_-$, where $G_-$ is the *ignition point*), local dynamics is poorly affected by the dynamics of the neighbouring regions in the connectome, and all regions are in low activity baseline state (hence $R_{max}$ is low) regardless of the ICs. Conversely, when $G$ is too high ($G>G_+$, where $G_+$ is the *flaring point*), the local dynamics are totally determined by exceedingly strong long-range inputs, and any IC results in a high activity ignited state (hence $R_{max}$ is high). When $G$ is intermediate, a complex interplay between local dynamics and long-range influences gives rise to more complex dynamics. For $G_-<G<G_+$, we observe that Low ICs result in a global network state in which all regions are not ignited (*baseline network state*), while High ICs originate a second global network state in which there is a mixture of regions with low baseline activity and regions with ignited activity (*ignited network state*). This collective network bistability (visualized by the existence of two branches of $R_{max}$ in the $G_-<G<G_+$ range between the ignition and the flaring points) results from the network dynamics, because the isolated nodes, with the parameters we employ here, display a single activation state. The actual fraction of ignited nodes, $F_{ignited}$, in the ignited network state depends on the chosen $G$ value in the bistable range and gradually increases from a minimum value $F_{ignited} = F_-$, observed at $G = G_-$, to a value $F_{ignited} = F_+ \sim 90\%$, observed at $G = G_+$.

See Materials and Methods for more details on model implementation and on determining the existence ranges for the baseline and ignited network states. In the next sections, we explore which features of the connectome included in the model determines the occurrence of the $G_-$ and $G_+$ points, as well as the increased profile of $F_{ignited}$ through the bistable range.

## The existence of a bistable ignition range does not depend on the human connectome topology

To test the relevance of the Human connectome (*Human*) organisation in determining the ignition behaviour, we compared the simulated dynamics of a mean-field model based on the *Human*, with alternative surrogate connectomes. The surrogate connectomes conserve key features of *Human* organisation while selectively randomising others.

We first considered *unweighted surrogate connectomes* (uSCs, Fig 2A) in which all connection weights were set to a uniform strength (equal to the mean value of the *Human*). In this way, we could disentangle effects on the collective network ignition dynamics that were genuinely due to the connectivity structure, irrespectively of the influence of the weight of the connections. A first surrogate connectome is the *Human$_{hw}$* whose connectivity pattern is identical to the *Human* reference but has homogeneous weight (Fig 2A, left; see Fig B in S1 File). We then considered an ensemble of unweighted Degree-Preserving Random (*DPR$_{hw}$*) surrogate connectomes, in which, in addition to making all the weights homogeneous (as in the *Human$_{hw}$* case), connections were randomly rewired between nodes by still preserving the degrees of each cortical area in the *Human* [32], as a signature of its local organisation (Fig 2A, middle; see Fig C in S1 File and Materials and Methods for details). Finally, we generated an ensemble of Small-World (*SW$_{hw}$*) surrogate connectomes optimized to conserve the global small-worldness [33,34] of the *Human* as a signature of its global organisation (Fig 2A, right). The small-worldness values of *SW$_{hw}$* are close to *Human* (see Fig D in S1 File and Materials and Methods for details) but the specific degree of the nodes are disrupted. Simulations of mean-field models embedding this uSCs allow probing whether the global small-worldness (for the *SW$_{hw}$* ensemble), the distribution of the local degrees (for the *DPR$_{hw}$* ensemble) or an

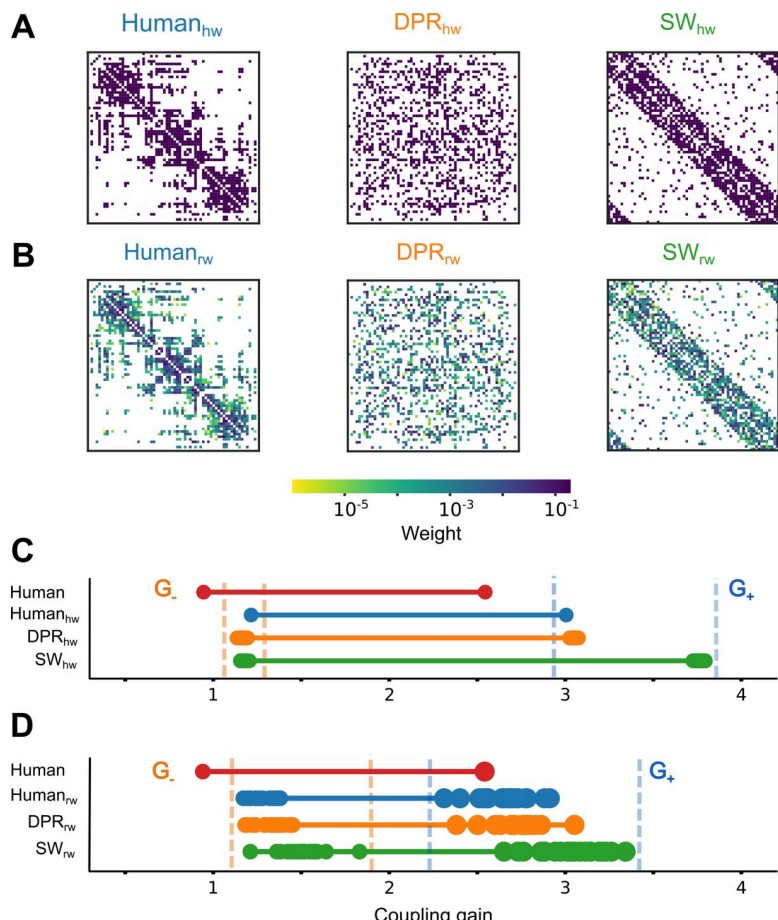

**Fig 2. The human cortical connectome requires a lower coupling gain to display ignition than surrogate models.**
**(A)** One example of the unweighted surrogate connectomes (uSCs) matrices, in which connections were normalized to make reliable comparisons with *Human* (each *purple* entry was set to $1.332 \times 10^{-2}$). The $DPR_{hw}$ networks disrupt the connectivity pattern but preserve the degree distribution. The $SW_{hw}$ networks display a small-worldness value close to the *Human*. **(B)** One example of the weighted surrogate connectomes (wSCs) matrices. The colour bar shows the connection weights in a log-scale. **(C-D)** Bistable range of the *Human* compared to either uSCs **(C)** or wSCs **(D)**, highlighting the bifurcation $G_-$ (left) and $G_+$ (right) points. The *orange* dashes (*blue* dashes) show the range of values for $G_-$ ($G_+$) in surrogate connectomes.

exact adjacency structure (for the $Human_{hw}$), but not the exact distribution of weights are essential or not in determining the observed ignition behaviour.

To assess the relevance of the weighted structure and detailed degree-to-weight correlations, we constructed additional *weighed surrogate connectomes* (wSCs, Fig 2B). These new ensembles were obtained from corresponding unweighted ones by keeping the same connectivity structure and reassigning the individual weight values of different links in the reference *Human* to randomly selected links in the surrogate connectomes. We thus generated an ensemble of weight-permuted Human ($Human_{rw}$) surrogate connectomes in which connectivity and weight distribution are identical to the reference *Human*, but weights randomly permuted across the different links (Fig 2B, left). Analogously, we generated weighted versions of the Degree-Preserving Random ($DPR_{rw}$) (Fig 2B, middle) and Small-World ($SW_{rw}$) (Fig 2B, right) surrogate connectome ensembles. All these three different ensembles maintain by construction the same weight distribution of the original *Human*, but the eventual weight-to-degree correlations were disrupted (See Fig E in S1 File).

We performed simulations of mean-field models embedding surrogate connectomes of all the uSCs (Fig 2C) and wSCs (Fig 2D) types and determined for each of the simulated models whether a network bistability range existed or not, and which was its extension, i.e. the values for the ignition point $G_-$ (orange dashes) and flaring point $G_+$ (pale blue dashes). We then compared these critical point values to the range found for the *Human* (shown in red). Remarkably we found that, independently from the chosen uSC or wSC surrogate ensemble, network bistability is always present in a range of coupling gain. The existence of a bistable ignition range is thus not unique to connectomes of the *Human* type.

## The human connectome has an exceptionally low ignition point

While all tested surrogate connectomes give rise to a bistable ignition range, the actual values of the ignition point $G_-$ and of the flaring point $G_+$ depend on the used surrogate ensemble.

Compared to uSCs, the reference *Human* connectome has the lowest values for both the bifurcation points $G_-$ and $G_+$ (Fig 2C). In other words, the *Human* connectome needs a lower excitatory strength to first trigger ignition in some region (at the ignition point $G_-$) and then to lose the low activity state (at the flaring point $G_+$). For each of the considered uSC ensembles, the values of $G_-$ and $G_+$ varied very little across different random instances from the same ensemble (i.e. small dispersion). The three $Human_{hw}$, $DPR_{hw}$ and $SW_{hw}$ surrogate ensembles have all closely matching values of the ignition point $G_-$, systematically larger than for the reference *Human* connectome. The $SW_{hw}$ surrogate ensemble has the largest average flaring point $G_+$. Thus, neither the degree distribution (shared with the $DPR_{hw}$), the small-worldness (shared with the $SW_{hw}$) or even the actual connectivity pattern (shared with the $Human_{hw}$) can alone account for the exceptionally low values of $G_-$ and $G_+$ found for the *Human*.

To disentangle if these low values could be explained by heterogeneity in the weight of connections, we considered then simulations performed with wSCs (Fig 2D). For wSCs, the variability of $G_-$ and $G_+$ values across different random instances from the same ensemble was larger than for uSCs (i.e. high dispersion). This large variability already suggests that specific weight-to-connectivity arrangements can influence how low or high critical points are (see below for further analyses). Introducing heterogeneous weights, generally shifted the median flaring points $G_+$ toward lower values than for uSCs. The flaring point for the *Human* reference connectome falls now well within the fluctuation range of flaring points for the $Human_{rw}$ and $DPR_{rw}$ ensembles while flaring points for the $SW_{rw}$ ensemble continue to be larger (although smaller than for the $SW_{hw}$ ensemble). This suggests that the flaring point $G_+$ value observed for the *Human* connectome can be accounted for by its degree and weight distributions (shared with the $Human_{rw}$ and $DPR_{rw}$ ensembles, but not with the $SW_{rw}$ ensemble), rather than by its small-worldness (shared with the $SW_{rw}$ ensemble, but not with the $Human_{rw}$ and $DPR_{rw}$ ensembles).

Yet, none of the wSCs gives rise to such a low ignition point as for the *Human* connectome. This property of the *Human* connectome is thus exceptional, in the sense in which it is unlikely to arise by chance in the organisation of the studied surrogate ensembles.

## The human connectome has an exceptionally compact and strong core

Network topology is most frequently characterised in terms of a *local* organisation (Fig 3A, left), such as node degree or strengths, or in terms of a *global* organisation (Fig 3A, right), such as the whole network small-worldness. However, such measures are not sufficient to capture the highly heterogeneous interplays of the connectivity structure, given by specific patterns of weight correlations between subset of nodes that define characteristic *mesoscale* structures in the network: motifs, communities, cores, etc. [25,26,35]. We remark that the *Human* largely

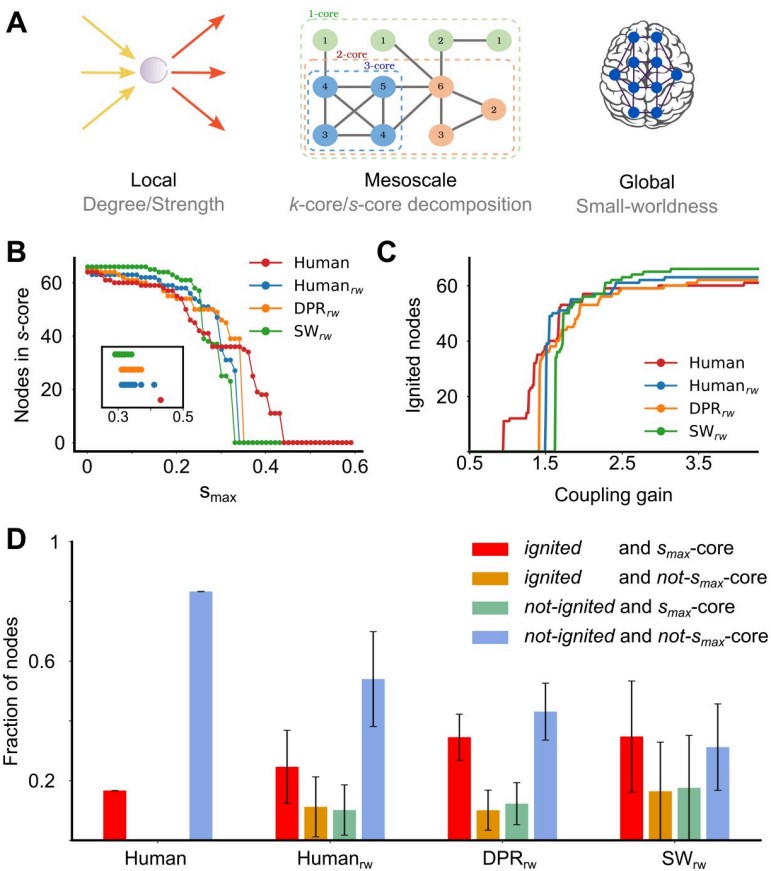

**Fig 3. Ignited cortical areas perfectly match the weighted core of the *Human* at the ignition point. (A)** A scheme of the local, mesoscale and global level of organization of the network. At left, the local level is represented by the sum of inputs and outputs of a cortical area. At the middle, the mesoscale level is measured with the core decomposition, composed by shells of incremental within-connected (or strongest) nodes. In the cartoon, *blue* nodes belong to the 3-core, *orange* to the 2-core shell and *green* to the 1-core shell. The number within each node denotes its degree. At the right, the global level considers the small-worldness, the integration and segregation ratio of the whole-network. **(B)** The *s*-core decomposition of $Human$ (*red*), $Human_{rw}$ (*blue*), $DPR_{rw}$ (*orange*) and $SW_{rw}$ (*green*). The *y-axis* shows the number of nodes in the shell, whereas the *x-axis* shows the $s_{max}$ of each shell. Only one example for each type of wSCs is shown in the main plot. The inset shows the $s_{max}$ for all the networks used. **(C)** Number of ignited ($R_i > 5$) nodes in the network as a function of coupling gain G and for high ICs, showing the *Human* and one example of each wSC. **(D)** Fraction of nodes which are *ignited* (red, orange) or *not-ignited* (green, blue) at the G- bifurcation, and that belong to $s_{max}$-core (red, green) or not (orange, blue).

shares local topological organisation with the *DPR* ensembles and global organisation with the *SW* ensembles, but none of these ensembles–in their unweighted or weighted versions–could account for the exceptionally low value of the *Human* ignition point. To chase for its eventual connectome-level determinants, we turn then to analyses of mesoscale structures and notably to the core-periphery organisation using the core-shell decomposition [36,37] of connectomes.

A network can contain subsets of nodes that are more strongly inter-linked between them than on average. Focusing first on unweighted graphs, we define as *k*-core a subgraph–i.e. a subset of nodes and the links interconnecting them–in which all the member nodes have at least *k* neighbours *within the subgraph*. The larger *k* is, the more difficult is to identify subgraphs that satisfy the *k*-core criteria, resulting in increasingly tighter cores. Any node member

of a $k$-core will also belong to any $k'$-core with $k' < k$, resulting in an "onion-like" nesting of progressively denser cores, up to a maximum value $k_{max}$ such that no $k$-core exists for any $k > k_{max}$ (cf. Fig 3A, middle; see Fig F in S1 File).

These definitions of cores and shells can be naturally generalized from unweighted to weighted networks by replacing the notion of node degree (discrete number of outgoing and ingoing connections) with the notion of node strength (sum of the continuous weights of outgoing and ingoing connections). Hence, an $s$-core is a subgraph such that all its nodes are connected between them with a strength larger or equal than $s$. There is a $s_{max}$-core, such that $s$-cores with $s > s_{max}$ do not exist anymore. In addition, one can define a smooth $s$-shell as a set of nodes belonging to $s'$-cores with $s < s' < s+\Delta s$ but not to the inner $s$-core (where $\Delta s$ sets a precision at which continuous $s$ values are quantized).

In Fig 3B we show the fraction of nodes belonging to $s$-core with increasingly larger $s_{max}$ for the *Human* (*red*), in comparison with representative instances from the different considered wSCs (see Fig G in S2 File for the analogous $k$-core decomposition). The *Human* contains a $s_{max}$-core with the largest $s_{max} = 0.431$ among all the other considered surrogate ensembles. The inset of Fig 3B portrays $s_{max}$ values for individual instances of the different surrogate ensembles showing that the *Human*'s $s_{max}$ value is larger than the $s_{max}$ of any individual instances as well. Furthermore, the $s_{max}$-core for the *Human* also includes a much smaller number of nodes ($n = 11$) with respect to the $s_{max}$-core of the other ensembles. Overall, thus, the *Human* has an exceptionally strong and compact $s_{max}$-core, that is unlikely to be found by chance in any of the tested *Human$_{rw}$*, *DPR$_{rw}$* and *SW$_{rw}$* surrogate ensembles.

In the case of our reference *Human* and in the adopted parcellation, the $s_{max}$-core included left and right Pericalcarine Cortex (PCAL), Cuneus (CUN), Precuneus (PCUN), Isthmus of the Cingulate Cortex (ISTC) and Posterior Cingulate Cortex (PC), as well as left Paracentral Lobule (PARC) (Table B in S1 File).

Remarkably, we obtained similar results with other two empirical connectomes, having different parcellations and network density, that display the same relationship (S3 File). Although the areas involved are different, they all contain a relatively compact $s_{max}$-core that is first ignited at a low value of global coupling strength. This core is also lost when the connectivity-to-weight pattern is disrupted by any randomization procedure.

## The human connectome core serves as an "ignition core"

As previously mentioned, the *Human* has the smallest ignition point $G_-$. We inspected in more detail which nodes get ignited when the ignited branch first appears at this low $G_-$. Fig 3C shows how many ignited nodes can be found in the ignited network state as a function of growing coupling strength $G$ for the *Human* and the different surrogate ensembles. We focused on the fraction $F_-$ of ignited nodes, observed at $G = G_-$. The *Human* has once again a particularly small fraction $F_-$ of early-ignited nodes, smaller than for any other surrogate ensemble (as visible by the large step jump occurring at $G_-$ in Fig 3C for the ignited fractions for the *Human$_{rw}$*, *DPR$_{rw}$* or *SW$_{rw}$* ensembles).

The actual number of ignited nodes at $G = G_-$ is $n = 11$, which is equal to the size of the compact *Human* $s_{max}$-core. As a matter of fact, this is not a coincidence. In Fig 3D we report the fraction of nodes that at the ignition point $G_-$ sustain an *ignited* state or that maintain on the contrary a baseline state (*not-ignited*), separating them further in nodes that belong or not to the $s_{max}$-core. All the nodes ignited at the ignition point $G_-$ also belong to the $s_{max}$-core in the case of the *Human*. Conversely, all the nodes in the $s_{max}$-core are ignited already at $G_-$. In other words, for the *Human*, the subset of regions that first sustain an ignited state at the critical ignition precisely match the $s_{max}$-core.

This one-to-one correspondence happens uniquely for the *Human* and is lost whenever connection weights are randomized (even when the connectivity pattern is maintained, as for *Human*$_{rw}$). In the case of the other surrogate connectomes, at the ignition point, there are always nodes ignited at G_ but not belonging to the $s_{max}$-core (*orange* bars in Fig 3D) or nodes belonging to the $s_{max}$-core but not ignited (*green* bars in Fig 3D). This "spill-over", present in all wSCs, is more pronounced for the $SW_{rw}$ ensemble.

The unusually large strength of internal connections within the *Human*'s $s_{max}$-core thus allows it to internally sustain ignited activity with a relatively weak strength of inter-regional coupling, explaining the smaller value of G_ for the *Human* (Fig 3C). Given that the inter-regional coupling is still weak, ignition does not propagate outside of the $s_{max}$-core but remains confined within it (Fig 3C). The fraction of early-ignited nodes also remains smaller than for other wSCs because the *Human*'s $s_{max}$-core is particularly small-sized. On the contrary, for other wSCs, a higher global strength of coupling is required to trigger ignition and therefore ignition can also immediately propagate beyond the core, resulting in larger fractions of early-ignited nodes.

Summarizing, the exceptionally strong $s_{max}$-core of the *Human* serves as an "ignition core", that highly facilitates ignition within a well precise and compact set of regions and allows then its active maintenance.

## Ignition is determined by *s*-coreness more than by other connectome features

For the *Human*, the set of first-ignited regions at the ignition point G_ is predicted by the $s_{max}$-core network feature. The regional propensity to early ignition is not however equally predicted by other network features. For instance, considering $k_{max}$-core, the unweighted analogue of $s_{max}$-core, near the 80% of the nodes (42 of 53) in the *Human*'s $k_{max}$-core are not ignited at the G_, as shown by Fig 4A (leftmost histogram, *green*). Analogously, several nodes with the degree (Fig 4B, leftmost) or strength (Fig 4C, leftmost) as high as one of the early-ignited nodes are not yet ignited at the ignition point G_. Thus, for the *Human*, none of the probed network features reaches the perfect prediction of early ignition achieved by the $s_{max}$-core.

Apparently, $k_{max}$-core predicts early ignition better for the other surrogate connectome ensembles than for the reference *Human*. However, this is a consequence of a larger fraction of nodes being ignited at G_ in the $DPR_{hw}$ and the $SW_{hw}$ ensembles. Still, for both these ensembles, some of the early ignited regions do not belong to the $k_{max}$-core (Fig 4A, right panels).

Overall, the analyses of Fig 4 confirm that the *Human* connectome is associated with exceptional connectivity-to-weight correlation patterns, responsible for its exceptional ignition behaviour.

## The human connectome supports an exceptionally graded cortical ignition dynamics

We then studied the order of ignition of additional regions when increasing the coupling strength above the initial ignition point G_, taking track of the actual value of the coupling in which they first become able to sustain ignition.

Fig 5A and 5B shows, for *Human* and *Human*$_{rw}$, the ranges of G over which different regions support an ignited state, ranking them from bottom to top in order of earliest ignition. In addition, we colour-coded the $s_{max}$ of the nodes (see Figs H-L in S2 File for analogous plots using other strength features and other wSCs). It is visually evident, from the inspection of Fig 5A, that the order of recruitment into the ignited state is closely correlated to the rank of $s_{max}$

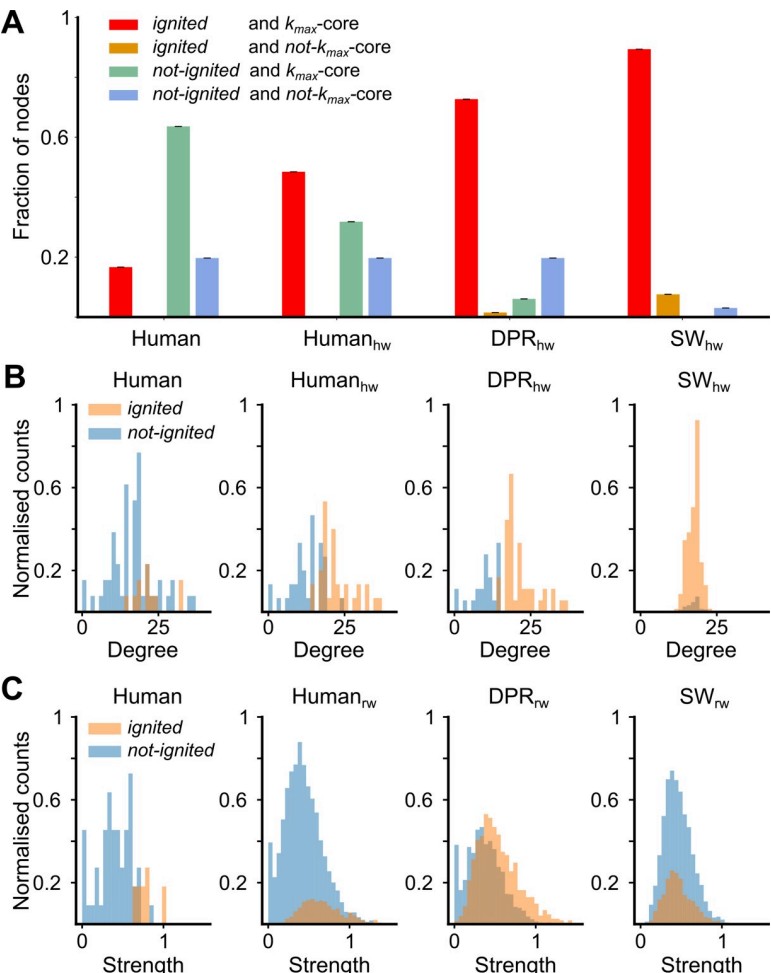

**Fig 4. Ignited cortical areas are loosely related to other organization features at the ignition point. (A)** Fraction of nodes which are *ignited* (*red*, *orange*) or *not-ignited* (*green*, *blue*) at $G_-$, and that belong to $k_{max}$-core (*red*, *green*) or not (*orange*, *blue*). Note that the $k_{max}$-core nodes match with all the *ignited* in the `Human`, but also with a large number of nodes with baseline activity (*not-ignited*). **(B)** Degree distribution of ignited (*orange*) and not-ignited (*blue*) nodes at the ignition point $G_-$, for `Human` and the wSCs. **(C)** Strength distribution of *ignited* (*orange*) and *not-ignited* (*blue*) nodes at the ignition point $G_-$, for *Human* and the wSCs.

values of the different regions. The colours in Fig 5B (*Human$_{rw}$*) are more disordered, indicating that this relationship is disrupted by random permutation of connection weights. This correlation is quantified in Fig 5C, where the explained variance of the Spearman correlation ($\rho^2$) between ignition and $s_{max}$ ranks to be as large as 0.88 (the lower the $s_{max}$, the later a region can support an ignited state). However, the $\rho^2$ between the rank of ignition and rank of total and in-strength drops to 0.67 and even down to 0.38 for the rank of out-strength.

In Fig 5D we show that none of the surrogate connectomes achieves such a large correlation between ranks of $s_{max}$ and ranks of ignition as the *Human* connectome. This is due to the fact that the *Human* connectome has the broadest distribution of $s_{max}$ values across the regions, resulting in a relatively smooth increase in the number of ignited regions with growing $G$. On the contrary, in surrogate connectomes–and particularly in the $SW_{rw}$ one–the gap between the largest and the smallest $s_{max}$ is narrower, leading to an abrupt increase of the number of ignited regions with growing $G$. Furthermore, in surrogate connectomes, we observe "spill-over" even beyond the early ignited subset of regions, i.e. frequent recruitments of regions with smaller $s$-

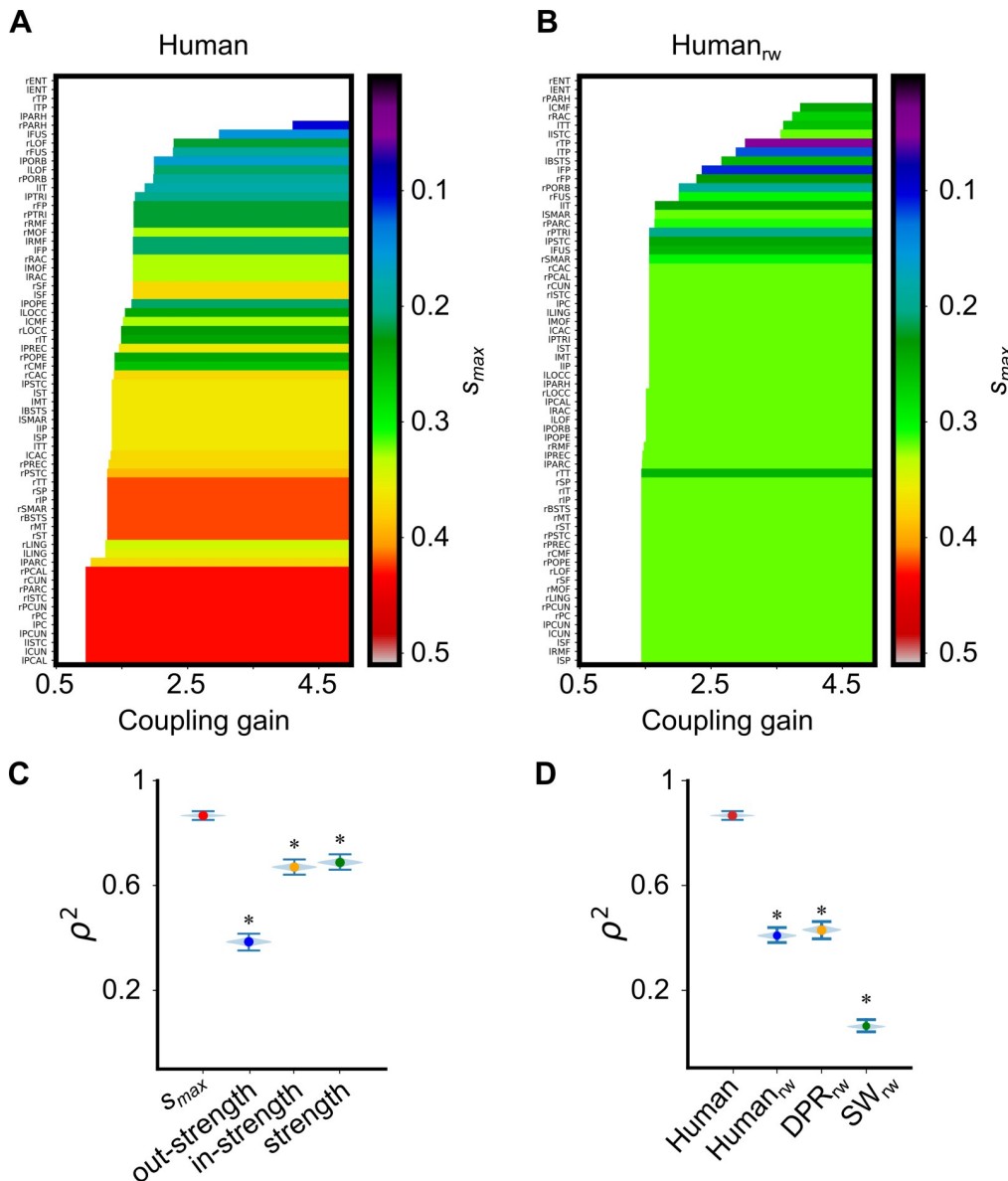

**Fig 5. The weighted core-shell organization of *Human* is more related to the growing of ignited nodes than other surrogate connectomes or organization levels. (A)** Cortical areas sorted in the *y-axis* according to the coupling gain G value at which they first ignite. The colour code shows the $s_{max}$ for each of the ignited cortical areas of *Human*. **(B)** The same for the *Human_{rw}*, to stress the difference in ignition recruitment through the core-shell organisation. **(C)** Spearman rank correlation variance ($\rho^2$) between first ignition value *G* and the $s_{max}$ (**0.867**, percentile (2.5, 97.5) = (0.858, 0.874)), out-strength (**0.386**, percentile (2.5, 97.5) = (0.369, 0.402)), in-strength (**0.670**, percentile (2.5, 97.5) = (0.655, 0.684)), and strength (**0.688**, percentile (2.5, 97.5) = (0.672, 0.703)) of a node. The * indicates a significative difference between $s_{max}$ and in-, out-, and strength. **(D)** $\rho^2$ between ignition value *G* and the $s_{max}$ for *Human*, *Human_{rw}* (**0.474**, percentile (2.5, 97.5) = (0.459, 0.490)), *DPR_{rw}* (**0.495**, percentile (2.5, 97.5) = (0.477, 0.512)), and *SW_{rw}* (**0.100**, percentile (2.5, 97.5) = (0.088, 0.112)). *Human* shows a higher explained variation by the core-shell organization than the wSCs. The * indicates a significant difference between the $\rho^2$ of *Human* and wSCs. The significance of $\rho^2$ was evaluated using 10.000 replicas from bootstrap resampling (violin plots).

coreness and failed recruitment of regions with higher $s_{max}$, confirmed by the decrease of the explained variance of the Spearman correlations in Fig 5D. Also, the explained variance of the ignition is higher in *Human* than in wSCs (Fig M in S2 File).

## Discussion

Our computational modelling investigations have shown that bistability of cortical activation can robustly and naturally occur in the resting-state as an effect of the interplay between regional dynamics and long-range interactions mediated by the cortical connectome. Not all the regions display the same propensity to get ignited. Eventually, via analyses of the graph topology of the connectome–and, notably, of its weighed core-shell structure–, we were able to predict with large accuracy the order into which different regions can get spontaneously ignited with increasing inter-regional coupling. We found that regions belonging to a maximally strong $s$-core are among the first to sustain spontaneous ignition during simulated resting state. Comparing the *Human* with a variety of random surrogate connectome ensembles, we found that empirically observed connectomes are "non-random", in the sense that they display an exceptionally strong and compact $s_{max}$-core and give rise to a particularly smooth and gradual increase in the number of ignitable regions as a function of the strength of inter-regional coupling.

The SC organisation is thus a strong determinant of the observed collective dynamics, in line with previous evidence [6,8,9,38]. More than local topology metrics, such as degrees or strengths, or global topology metrics, such as overall small-worldness, we found a *mesoscale* topological organisation, $s_{max}$ of the core, to be the best predictor of the bistable activity patterns expressed by the model. Most of the regions with largest $s_{max}$-core in our models, such as Cuneus, Cingulate, or Precuneus cortices are also members of what Hagmann et al. [27] dub the "*structural core*" of Human cerebral cortex, as well as strongly functionally implied in Default Mode Network fluctuations [39]. Such a set of densely interconnected regions had already been hypothesized to play an important role in shaping large-scale resting-state dynamics [27,38,40], a hypothesis which we here further confirm.

From a more abstract statistical mechanics perspective, coreness and core-shell decompositions had been used to describe the propagation of infection on complex networks with inhomogeneous density [37]. Here, an analogy could be drawn between "*ignition*" and "*infection*", with ignition being first possible in the densest $s$-cores, where nodes in a strongly connected neighbourhood can trigger each other into an ignited state by mutual excitation (analogously to infection) and mutually stabilize their ignited state by preventing the return to baseline state (analogously to suppressed recovery). Interestingly, the rank correlation between the order of ignition and the in- or out-strengths in the connectome for different regions were stronger for in-strengths than for out-strengths. This fact indicates that a core region that can be "infected" by its neighbours (i.e. triggered by them into an excited state) will be more likely to remain ignited than a region who can "infect" its neighbours via strong output connections. In line with this observation, some areas in the Human network never get ignited within the range of G that we explored. These areas correspond to the temporal pole (TP), entorhinal cortex (ENT) and the left parahippocampal cortex (lPARH), which are also the regions of lowest in-strength and at the most peripheral shells of the network.

The graded ignition is also reminiscent of the stretching of the criticality of cortical-network dynamics via Griffith phases [8]. In this scenario, critical-like dynamics emerges for a range of the control parameter rather than for a single point, due to the structural connectivity of a hierarchical modular network. In a Griffith phase, criticality emerges from the rare-regions, a subset of nodes with activity values significantly different from their system averages [8,9]. In our work, the bi-stable range starts at G₋ with the $s_{max}$-core regions, a small fraction of the network, evoking the rare-regions concept. Moreover, in the case of the uSCs and wSCs, the number of regions ignited at G₋ is larger than low activity regions. In the case of the $SW_{hw}$ ensembles, almost all the network nodes ignite, as predicted by the stretching of criticality

work of Moretti & Muñoz [8] (Fig N in S2 File). However, it remains difficult to speak rigorously of Griffiths phases for network systems having a small number of nodes like the ones we used in this study.

The precise regions that belong to the largest $s$-core of the connectome do vary depending on the specific chosen empirical reconstruction, and their enumeration is also necessarily affected by the used parcellation. In S3 File, we show indeed that, comparing two alternative empirical reconstructions of the human cortical connectome, the overlap between the included regions is only partial (Table B in S1 File). Remarkably, however, for all these alternative human connectomes the set of regions that are early ignited largely match the largest $s$-core. This is not true, on the contrary, for the considered surrogate connectomes: all of them display a higher degree of "ignition spill-over" (early ignited regions outside the largest $s$-core) or "incomplete ignition" (some of the regions in the largest $s$-core not igniting). It may be that the use of *ad hoc* search procedures (e.g. genetic algorithms [41]) will allow engineering non-standard surrogate connectomes which would display *Human*-like or even better than *Human* ignition capabilities. However, we failed to identify any obvious graph-theoretical feature that confers to *Human* $s$-cores their exceptional ignition boosting properties, beyond the ones of generic $s$-cores.

Finally, ignition dynamics is affected not uniquely by an individual graph-theoretical organisation of the connectome but by correlations between multiple properties as well. This fact is epitomized by the differences in ignition dynamics between the *Human* and *Human$_{rw}$* connectomes. Indeed, the *Human$_{rw}$* connectome shares with the *Human* identical unweighted topology and distribution of weights but the correlations between the two have been disrupted. Analogously, surrogate connectomes with randomized weights display a larger variability over the ensemble of the actual values of the ignition and flaring points $G_-$ and $G_+$ than unweighted ensembles. The fact that all instances within these surrogate ensembles with randomized weights share the same weight distribution and a common statistical distribution of degrees or other topological properties confirms that the critical ignition behaviour of the model is influenced much more by weight-to-topology correlations than by weights or topology independently.

Here we are describing in the connectome an organization that cannot be explained only in terms of pairwise node-to-node relationships. Interactions between more than a pair of nodes (high-order interdependencies) are described using information theory tools [42,43], giving rise to phenomena like redundancy and synergy that appear in the brain activity. These cannot be understood nor measured if only pairwise relationships are quantified. A similar situation can be occurring at the connectome level, and the $s$-core decomposition is a first step toward the description of this kind of structural high-order interactions. A question for future research is whether the functional high-order interactions–as the one revealed by non-trivial "meta-connectivity", constraining fluctuations of pairwise resting state functional links [44]–are related to the core-shell organization.

Even if we cannot yet fully explain the observed ignition behaviour of the model in terms of the network organisation of the connectome it embeds, these organisations remain nevertheless a strong determinant of the observed dynamics. This finding is in apparent contrast with theoretical works based on more abstract network topologies [10,11,17] in which the variety of possible dynamical behaviours transcends structural complexity. A first possible reason is that dynamical diversity is strongly amplified by connectome symmetries and the resulting possibility of a multiplicity of ways of breaking these symmetries [10]. Now, the *Human* connectome, with all its characteristic heterogeneities and idiosyncrasies, is far from being symmetric. Second, we focus in this work on the network multistability between the two main ignited and baseline activity branches of the mean-field whole brain model. However, other sub-dominant

states exist between the early ignition $G_-$ and the late flaring $G_+$ points, in which the spatial patterns of regional low or high activation levels are less influenced by the structural backbone [15]. Finally, we adopted here a very simple regional dynamics, with bistability between just two fixed points, but we expect that using neural masses able to express richer regimes–oscillatory, bursting, chaotic, etc. [17,45,46]–could eventually reduce the sway of connectivity on collective emergent dynamics.

Future extensions of our model will have not only to embed richer dynamics but also to investigate more dynamic notions of ignition. The specific way in which we treat ignition within the present study is rather static. We focus on the possibility that specific regions develop bistability between a baseline and an ignited state and we track at which value of the inert-regional coupling $G$ this bistability becomes first sustainable. However, we do not study the effects on the ongoing dynamics of an actual switching from baseline to ignited state occurring. Experimentally, local ignition is associated with a "glow", e.g. to a reverberation of enhanced activation followed by propagation toward neighbouring regions [20,21]. Recently, mean-field whole-brain models able to reproduce certain conditions such as propagation of ignition, thanks to a balanced amplification mechanism, have been introduced [24]. Analogously, other modelling studies have measured the "intrinsic ignition" capabilities of different regions by quantifying their capacity to propagate to neighbouring regions the effects on activation of a locally received perturbation [22]. In our model, we expect that, near the ignition point, perturbing a node within the largest $s$-core to switch from baseline to a locally ignited state would quickly result in all the other nodes within the largest $s$-core to get ignited as well, given the strong mutual excitation loops present within this core. However, we chose here for simplicity to characterize the collective equilibrium state after network ignition has taken place, postponing to future studies the investigation of the out-of-equilibria transient dynamics leading to these ignited equilibria. In this sense, our static definition of an ignition core as the subset of regions whose local dynamics is pushed by network dynamics to be close to its critical instability point–making them able to easily switch between low and high firing rate states–is quite related to the notion of "dynamic core" introduced by Deco et al. [47]. Dynamic core regions, indeed, identified after the convergence of a fitting procedure (and not by the study of their participation into ignition dynamic transients), are defined as sitting closely at the bifurcation between asynchronous and oscillating local states.

Even without studying the actual propagation dynamics of ignition, our modelling approach discovered that the effects of ignition (i.e. the resulting ignited network states) supported by the *Human* connectome are the most graded and fine-tunable among all the tested surrogate connectomes. In the ignition framework of Deco and Kringelbach [22] four classes of ignition are defined, that range from weak non-hierarchy to graded uniform hierarchy (Fig 6A–6D). In the first case, all the nodes have the same susceptibility to be ignited, while in the latter case there exists a linear uniform gradation in the ignition of the nodes. Between these poles, two other classes are staircase hierarchy and graded non-uniform hierarchy. Our results fit better with the staircase hierarchy class; there is a subset of nodes susceptible to be ignited and this number is smoothly controlled by the coupling gain (Fig 6E, orange arrow). In the randomized networks, there is a narrower range for the recruitment of cortical areas as G is increased (Fig 6F). Moreover, our investigation of randomized surrogate connectomes reveals that the likelihood that connectome structures supporting such a smooth hierarchy of possible ignited network states arise by chance is rather small. Thus, there must be some reason for which the *Human* happens to be as it is, a needle in the haystack of possible connectomes.

A first scenario is that the selection of a connectome with such non-random features is driven by *developmental* constraints, imposing specific construction principles to be respected but keeping network connectivity otherwise maximally random. Rubinov [48] evokes the

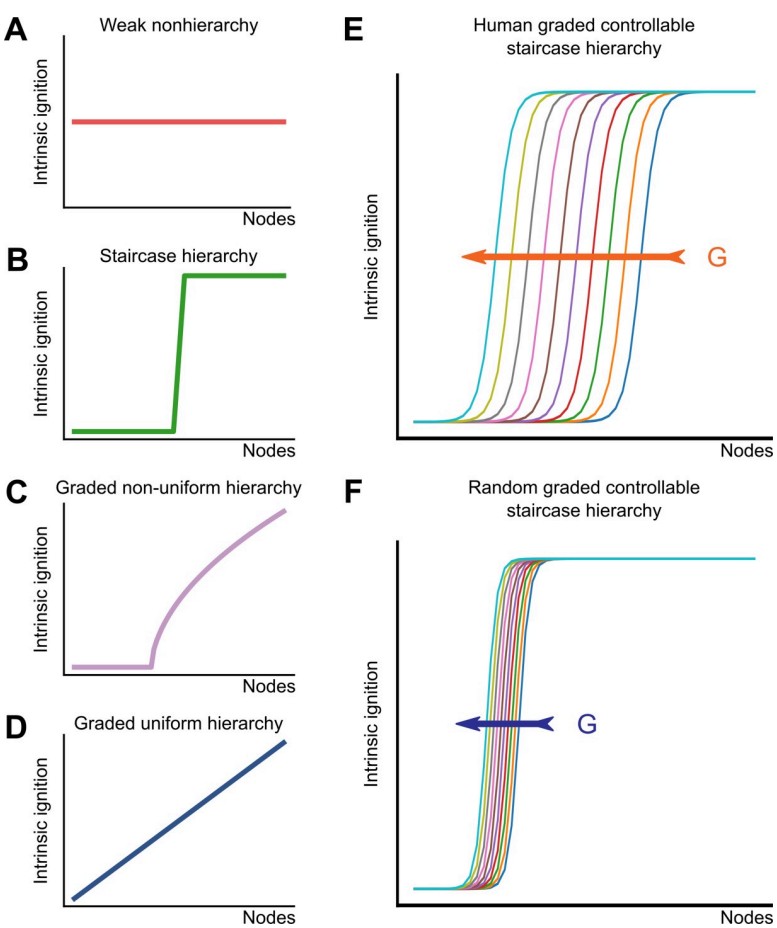

**Fig 6. The intrinsic ignition framework.** Deco and Kringelbach [22] define four classes of network ignition, that range from **(A)** weak nonhierarchy to **(D)** graded uniform hierarchy. Between these poles, two other classes are **(B)** staircase hierarchy and **(C)** graded non-uniform hierarchy. **(E)** In the *Human* connectome, the number of nodes susceptible to be ignited is smoothly controlled by the coupling gain, as shown in the orange arrow. In the **(F)** $DPR_{rw}$ the number of ignited nodes is less controllable.

notion of "*spandrel*", the triangular spaces that are unavoidably created between arches, pillars and beams when constructing a cathedral. These spandrels are statistically as frequent than the other structural architectural elements–the arches, pillars and beams that bear the weight of the building–but are not in the plan, i.e. they are byproducts of other constraints and construction targets. Such a scenario of the emergence of a *Human*-like ignition-core as a byproduct of some other graph-theoretical construction rule, e.g. imposed degree or small-worldness, was implicitly probed by our procedure of testing the *Human* connectome against null-hypotheses, represented by increasingly more constrained families of surrogate connectomes. Our failure to reproduce *Human*-like ignition-cores in any of the attempted surrogates leaves however open the question of which could be the hidden developmental constraints inducing the emergence of the exceptional *Human* s-core.

A second scenario is that such an exceptional $s_{max}$-core as the *Human*'s does not emerge as a "spandrel" but is actually favoured over others along with evolution for the fitness, if not optimality in some sense, that it confers. Interestingly, empirical connectomes extracted from another non-human organism [40,49,50], also include prominent structural cores in their organisation that match the set of firstly ignited nodes (S3 File). Future investigations may

check whether an ignition behaviour as the one we observed for *Human* connectomes is progressively set in place while adopting connectomes that follow a phylogenetic sequence, even if comparative connectomic analyses are still incomplete [40,51]. And, yet, the specific optimization goals with respect to which the empirical connectomes should be constructed are unknown. Several independent studies suggest that wiring cost minimization may be relevant but not sufficient to explain the observed connectome wiring, that at the same time seems to optimize information-processing related quantities [52,53].

Here we advance the hypothesis that the eventual reasons making the empirically observed connectome fit, and thus selected under evolutionary pressure, could (speculatively) be: first, the exceptionally low ignition point $G_-$ that it confers, allowing to initiate and maintain an ignited state with relatively low inter-areal couplings (and thus more limited use of synaptic transmission resources and connecting fibres amount); and, second, the exceptionally graded increase of the number of regions admitting bistable ignition when further increasing the inter-regional coupling $G$. Indeed, thanks to this graded rise, changes in the cortical networks' "*working point*" induced, e.g. by neuromodulation [54,55], arousal or other intrinsic or extrinsic mechanisms, here phenomenologically modelled by changes of the effective $G$, would give rise to the largest possible variety of possible ignition patterns and therefore, possibly, to subtle controllability of the extent of inter-regional integration. Our hypothesis implicitly postulates a positive functional role for the existence of subsets of ignited regions and the possibility of their fine-tuned control (Fig 6). As previously mentioned, the emergence of ignited activity into extended regional subsets, beyond early sensory areas has been repeatedly associated to aware perception [20], requiring recruitment of a global workspace [56]. In this sense, connectomes facilitating early ignition would favour at the same time, the emergence of a substrate dynamical repertoire required for integrated perception and, more in general, integrated information processing. Analogously, the possibility of supporting a graded hierarchy of possible ignited network states, recruiting narrower or wider nested circles of regions, could provide the mechanistic basis for "*graded consciousness*" states [57], in which workspace ignition can take place in a variety of possible ways, encompassing an increasing number of possible dimensions [58], rather than just being "*all-or-none*".

## Methods

### Structural connectomes

**Human cortical connectome.**   We used the human cortical connectome derived from *diffusion* MRI provided by [27], which corresponds to an average of five right-handed male subjects. This SC has 66 cortical areas, defined by a standard parcellation scheme provided by FreeSurfer [31] and 1148 connections determined by the DSI analysis (Fig 1B). The connection weights are normalised by the number of tracts and relative volume among two cortical areas (details in [27]).

**Surrogate connectomes.**   To make valid comparisons with the human connectome, we used surrogate connectomes that disentangle either unweighted or weighted network properties [59].

*Unweighted Surrogate Connectomes (uSCs).* To study how topological network features impact the dynamics of human connectome (*Human*), we homogenised the connection weights making them equal to the mean of *Human*. In other words, each connection was set to $1.332 \times 10^{-2}$. We made a homogeneous weight version of *Human* that preserves its connectivity pattern, $Human_{hw}$. Also, we built 100 equivalent Degree-Preserving Random ($DPR_{rw}$) connectomes with the Maslov and Sneppen algorithm [32]. The $DPR_{rw}$ maintain the number of nodes and edges, as well as the degree distribution of the *Human* [48,60–62]. Finally, the Watts and Strogatz Small-World model was used to generate 100 connectomes ($Human_{hw}$) which

maintain globally distributed processing and regional specialization of the *Human* [33,34,62–64]. First, we built 1000 $SW_{hw}$ networks and then selected 100 that had the most similar value of the *small-world coefficient*, σ, of the *Human*.

*Weighted Surrogate Connectomes (wSCs).* In the weighted Surrogate Connectomes (wSCs), we used the uSCs and randomly assigned to their connections the weights of the *Human*. In this way, we preserved the connection weight distribution of the *Human* [61]. With this procedure, we made 60 *Human_{rw}*, 60 *DPR_{rw}*, and 60 *Human_{rw}* networks ('rw' stands for random weights).

## Network metrics

We split the topological organisation of the networks in local, mesoscale and global to assess their correspondence with activity states of the nodes. To identify nodes that are locally relevant in the network, we used the degree $k(i)$, and strength $s(i)$ measures. Degree quantifies the number of links that directly connect to node $i$, whereas the strength is the sum of the weighted inputs and outputs to a node $i$ in the network [48,65].

In a similar manner, we used the core-periphery organisation as a mesoscale feature of the networks [66]. To identify the core of densely interconnected nodes in the network, we used the *k-core decomposition*, in which the shell of nodes with degree $<k$ are recursively removed to obtain its core nodes [37,64,67]. Therefore, the maximal $k$-core ($k_{max}$-core) defines the largest $k$ value, at which a highly interconnected sub-network exists. Similarly, the *s-core decomposition* defines the core of interconnected nodes with strength $s$ or higher among them. Thus, maximal $s$-core ($s_{max}$-core) is the more strongly inner-connected core of the network [15,27].

Finally, we used the *small-world index*, σ, as a global organisation that reveals the balance between high clustering of the nodes and the short-path length of connections between nodes. The small-world index is the ratio between the normalised clustering coefficient, γ, (fraction of node neighbours that also connect with each other) and the normalised characteristic path length, λ (shortest average path-length between nodes) of the network [48,63,65,68]. A network has the small-world property when σ > 1 [34,62,63]. An equivalent random network is used to normalise λ and γ.

$$\sigma = \frac{\gamma/\gamma_{random}}{\lambda/\lambda_{random}}$$

We found that σ of the *Human* is 1.63 (+/- 4.3x10$^{-3}$), its $\lambda/\lambda_{random}$ is 1.07 (+/- 5.4x10$^{-4}$), and its $\gamma/\gamma_{random}$ is 1.74 (+/- 4.6x10$^{-3}$). Thus, the *Human* has the small-world property.

## Dynamical mean-field model

We used the Wong-Wang mean-field model to simulate the local dynamics of each cortical area, which is a sum of self-recurrent activity, the network inputs, and the basal activity [15,19,69]. We implemented the deterministic version of the model to observe the attractor structure of the collective dynamics.

$$\frac{dS_i}{dt} = -\frac{S_i}{\tau_s} + (1 - S_i)\gamma R_i$$

$$R_i = \frac{a\chi_i - b}{1 - \exp(-d(a\chi_i - b))}$$

$$\chi_i = wJ_N S_i + J_N G \sum_j C_{ij} S_j + I_0$$

$S_i$ represents the open fraction of NMDA channels, $R_i$ is the mean firing rate, and $\chi_i$ represents the total synaptic input, all of them for the $i$th cortical area. We systematically explored the parameter of the coupling gain, $G$. $C_{ij}$ is the SC matrix with the connections from node $j$ to node $i$. Values for the other parameters are $\tau_s$ = 100 ms, $\gamma$ = 0.641, a = 270 $(\text{V}\cdot\text{nC})^{-1}$, b = 108 Hz, d = 0.154 s, $\omega$ = 0.9, $J_N$ = 0.2609 nA and $I_0$ = 0.3 nA. Each simulation was run for 120 $s$ with time steps of $\Delta$t = 1 ms, using a Euler integration scheme. Simulations were run with scripts written in Python.

## Computer simulations and fixed-point analysis

To describe the collective states of the networks, we used the fixed-point analysis of the stationary dynamics [15,23]. The parameter $G$ was varied in the range of $0.5 \leq G \leq 5$, with steps $\Delta G$ = 0.01. For each $G$ value, the simulations were run using random ICs drawn from a uniform distribution in one of two ranges: High ICs ($0.3 \leq S_i \leq 1$) or Low ICs ($0 \leq S_i \leq 0.1$). In this work, we modified the range of ICs described by [15,23] to ensure that the bistability range is the broadest possible. Then, we took the highest value of the $R_i$ activity among all nodes, denoted $\boldsymbol{R_{max}}$, to indirectly capture the network activity state. The bifurcations $G_-$ and $G_+$ are defined as the minimum and maximum $G$ value for which the $R_{max}$ depends on the ICs. Below $G_-$, simulations always finish in the same (low) $R_{max}$, regardless of the ICs; similarly occurs for gain coupling above $G_+$ with a high $R_{max}$. Although ICs were randomly chosen, the bifurcations $G_-$ and $G_+$ for a given network always had the same value, and this was checked by running 60 simulations for each combination of $G$, ICs and network.

## Thresholding of node activity

We used a threshold of activity to classify the nodes in high activity or low activity. After examination of the typical values of $R_i$ in the simulations, we established a threshold of $R_i > 5$ to assign a node to the *high firing rate* (*ignited*) subset; otherwise, they are part of *low firing rate* (*not-ignited*).

## Network tools

Structural models and network analyses used in this paper were carried out using the Python modules *bctpy* and *brainconn*, both python implementations of the publicly available Brain Connectivity Toolbox [48].

## Supporting information

**S1 File.** Supplemental Figures A-F and Supplemental Tables A-B.
(PDF)

**S2 File.** Supplemental Figures G-N.
(PDF)

**S3 File. Ignition, *s*-core and strength in alternative empirical connectomes.**
(PDF)

## Acknowledgments

We thank Enrique C.A. Hansen for scientific discussions and Antonio Roque for hospitality during the LASCON computational neuroscience school, during which the first steps toward this project were undertaken.

## Author Contributions

**Conceptualization:** Samy Castro, Wael El-Deredy, Demian Battaglia, Patricio Orio.

**Data curation:** Samy Castro, Demian Battaglia.

**Formal analysis:** Samy Castro, Demian Battaglia, Patricio Orio.

**Funding acquisition:** Samy Castro, Wael El-Deredy, Demian Battaglia, Patricio Orio.

**Investigation:** Samy Castro.

**Methodology:** Samy Castro, Wael El-Deredy, Demian Battaglia, Patricio Orio.

**Project administration:** Patricio Orio.

**Resources:** Samy Castro, Patricio Orio.

**Software:** Samy Castro, Patricio Orio.

**Supervision:** Wael El-Deredy, Demian Battaglia, Patricio Orio.

**Visualization:** Samy Castro, Patricio Orio.

**Writing – original draft:** Samy Castro, Demian Battaglia, Patricio Orio.

**Writing – review & editing:** Samy Castro, Wael El-Deredy, Demian Battaglia, Patricio Orio.

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
