## [Decision Letter · Decision Letter 0]

24 Apr 2020

Dear Mr. Orio,

Thank you very much for submitting your manuscript "Cortical ignition dynamics is tightly linked to the core organisation of the human connectome" for consideration at PLOS Computational Biology. As with all papers reviewed by the journal, your manuscript was reviewed by members of the editorial board and by several independent reviewers. The reviewers appreciated the attention to an important topic. Based on the reviews, we are likely to accept this manuscript for publication, providing that you modify the manuscript according to the review recommendations.

Sincerely,

Peter Neal Taylor

Associate Editor

PLOS Computational Biology

Daniele Marinazzo

Deputy Editor

PLOS Computational Biology

[LINK]

Reviewer's Responses to Questions

**Comments to the Authors:**

Reviewer #1: In this work, Castro and colleagues explore the factors favouring intrinsic ignition in brain activity using a mean-field whole-brain modelling approach by varying the strength of effective inter-regional coupling gain. In particular, they investigate the propensity of the human structural connectome to sustain intrinsic ignition and compare it to a variety of different random connectivity matrices. The authors show that the existence of a bistable ignition range does not depend on the connectome topology, but the human connectome reaches the ignition point for lower inter-regional coupling gain, which the authors link to its compact and strong core, given that these are the first regions to be ignited.

The paper is very well written, with the introduction covering the most relevant literature and the discussion covering the most important points and scenarios. The choice of the model is adequately justified to test the hypothesis in question and the authors demonstrate effort in making clear and illustrative Figures of the methods and results. Although the paper is publishable in its current form, I have a few suggestions to improve the quality of the manuscript.

Comments/Corrections/Suggestions:

Abstract:

weighed -> weighted

Last word of the abstract: dynamic -> dynamics (dynamic is an adjective, dynamics is the noun)

Summary:

Corresponds with -> corresponds to

Introduction:

Growing […] evidence stress -> stresses

The authors describe the results of the work directly in the introduction (page 5 and 6), which is not common practice. I suggest moving this to the discussion section and keeping the introduction to introduce the hypothesis that will be verified by the model. On the other hand, in the results section, the authors comment on the facts justifying the hypothesis, which could instead be moved to the introduction, i.e.:

in page 8: ‘When moving from regional to whole-brain network dynamics, we can expect, in agreement with several authors [6,13,18,26,27] that the spontaneous ignition dynamics and state shifting in different regions will be shaped by the underlying structural connectivity (SC) included in the model.’

In page 10: ‘The human connectome is associated with specific distributions of the local

organisation, such as node degrees (i.e. the number of neighbouring regions) or node in- or

out-strengths (i.e. the sum of the weights of incoming or outgoing connections), as well as of global organisation such as small-worldness [29,30]. It is not clear a priori how these different specific levels of organisation of the connectome influence the ignition behaviour of mean field models built on them. Therefore, to test the relevance of the Human connectome (Human) organisation in determining the ignition behaviour, we compared the simulated dynamics of a mean-field model based on the Human, with alternative surrogate connectomes. The surrogate connectomes conserve key features of Human organisation while selectively randomising others.’

Results:

In Figure 2C, the left blue dashed line (indicating the smallest G+) is not including the G+ obtained for the Human SC (which is lower). It is not entirely clear if this is a typo or if the authors intend to highlight only the range of G+ for non-human matrices. In any case, since the orange dashed lines G- include the Human SC values, it is important to verify this for consistency.

Page 8: ‘These heterogeneous activation levels could be distinguished into a low and high activation ranges’ -> into low or high activation ranges

Page 11: ‘Intriguingly, the actual number of ignited nodes at G = G- is n = 11, which is equal to the size of the compact Human smax-core.’

This finding seems quite expected rather than intriguing, so I suggest rephrasing the first word. At least to me, it appears more intriguing at first that the other structures do not show this, but it is of course, plausible.

In Figure 5 the ABCD labels are disproportionally large. I suggest reducing their font size.

Discussion:

In the discussion I suggest commenting on the fact that some areas never shown ignition in the region of bistability, such as the TP (temporal pole?) which is probably due to the fact that these areas are more distant to the structural core. Also, the way the SC is defined may increase the detection of connections at the core and decrease more distant ones.

Methods:

In the equations of the mean field model, there is a typo in the Ri equation (exp exp).

Reviewer #2: Uploaded as an attachment

**Have all data underlying the figures and results presented in the manuscript been provided?**

Reviewer #1: Yes

Reviewer #2: None

PLOS authors have the option to publish the peer review history of their article (what does this mean?). If published, this will include your full peer review and any attached files.

Reviewer #1: Yes: Joana Cabral

Reviewer #2: No
---

## [Decision Letter · Decision Letter 1]

15 Jun 2020

Dear Mr. Orio,

We are pleased to inform you that your manuscript 'Cortical ignition dynamics is tightly linked to the core organisation of the human connectome' has been provisionally accepted for publication in PLOS Computational Biology.

Best regards,

Peter Neal Taylor

Associate Editor

PLOS Computational Biology

Daniele Marinazzo

Deputy Editor

PLOS Computational Biology

Reviewer's Responses to Questions

**Comments to the Authors:**

Reviewer #1: The authors have adequately addressed my concerns and I recommend the paper for publication in PLOS CB.

Reviewer #2: The paper was already nice, and now it has been improved following all my suggestions as well as those of the other referee.

It can be accepted for publication.

**Have all data underlying the figures and results presented in the manuscript been provided?**

Reviewer #1: Yes

Reviewer #2: Yes

PLOS authors have the option to publish the peer review history of their article (what does this mean?). If published, this will include your full peer review and any attached files.

Reviewer #1: Yes: Joana Cabral

Reviewer #2: Yes: Miguel A. Muñoz

---

## [Editor Report · Acceptance letter]

23 Jul 2020

PCOMPBIOL-D-20-00105R1 

Cortical ignition dynamics is tightly linked to the core organisation of the human connectome

Dear Dr Orio,

I am pleased to inform you that your manuscript has been formally accepted for publication in PLOS Computational Biology. Your manuscript is now with our production department and you will be notified of the publication date in due course.

With kind regards,

Laura Mallard
